# Learning to Approximate a Bregman Divergence

Ali Siahkamari[1]    Xide Xia[2]    Venkatesh Saligrama[1]    David Castañón[1]    Brian Kulis[1,2]

[1] Department of Electrical and Computer Engineering
[2] Department of Computer Science
Boston University
Boston, MA, 02215
{siaa, xidexia, srv, dac, bkulis}@bu.edu

## Abstract

Bregman divergences generalize measures such as the squared Euclidean distance and the KL divergence, and arise throughout many areas of machine learning. In this paper, we focus on the problem of approximating an arbitrary Bregman divergence from supervision, and we provide a well-principled approach to analyzing such approximations. We develop a formulation and algorithm for learning arbitrary Bregman divergences based on approximating their underlying convex generating function via a piecewise linear function. We provide theoretical approximation bounds using our parameterization and show that the generalization error $O_p(m^{-1/2})$ for metric learning using our framework matches the known generalization error in the strictly less general Mahalanobis metric learning setting. We further demonstrate empirically that our method performs well in comparison to existing metric learning methods, particularly for clustering and ranking problems.

## 1   Introduction

Bregman divergences arise frequently in machine learning. They play an important role in clustering [3] and optimization [7], and specific Bregman divergences such as the KL divergence and squared Euclidean distance are fundamental in many areas. Many learning problems require divergences other than Euclidean distances—for instance, when requiring a divergence between two distributions—and Bregman divergences are natural in such settings. The goal of this paper is to provide a well-principled framework for learning an arbitrary Bregman divergence from supervision. Such Bregman divergences can then be utilized in downstream tasks such as clustering, similarity search, and ranking.

A Bregman divergence [7] $D_\phi : \mathbb{X} \times \mathbb{X} \to \mathbb{R}_+$ is parametrized by a strictly convex function $\phi : \mathbb{X} \to \mathbb{R}$ such that the divergence of $\boldsymbol{x}_1$ from $\boldsymbol{x}_2$ is defined as the approximation error of the linear approximation of $\phi(\boldsymbol{x}_1)$ from $\boldsymbol{x}_2$, i.e. $D_\phi(\boldsymbol{x}_1, \boldsymbol{x}_2) = \phi(\boldsymbol{x}_1) - \phi(\boldsymbol{x}_2) - \nabla\phi(\boldsymbol{x}_2)^T(\boldsymbol{x}_1 - \boldsymbol{x}_2)$. A significant challenge when attempting to learn an arbitrary Bregman divergences is how to appropriately parameterize the class of convex functions; in our work, we choose to parameterize $\phi$ via piecewise linear functions of the form $h(\boldsymbol{x}) = \max_{k \in [K]} \boldsymbol{a}_k^T \boldsymbol{x} + b_k$, where $[K]$ denotes the set $\{1, \ldots, K\}$  (see the left plot of Figure 1 for an example). As we discuss later, such *max-affine* functions can be shown to approximate arbitrary convex functions via precise bounds. Furthermore we prove that the gradient of these functions can approximate the gradient of the convex function that they are approximating, making it a suitable choice for approximating arbitrary Bregman divergences.

The key application of our results is a generalization of the Mahalanobis metric learning problem to non-linear metrics. Metric learning is the task of learning a distance metric from supervised data such that the learned metric is tailored to a given task. The training data for a metric learning algorithm is typically either relative comparisons ($A$ is more similar to $B$ than to $C$) [19, 24, 26] or

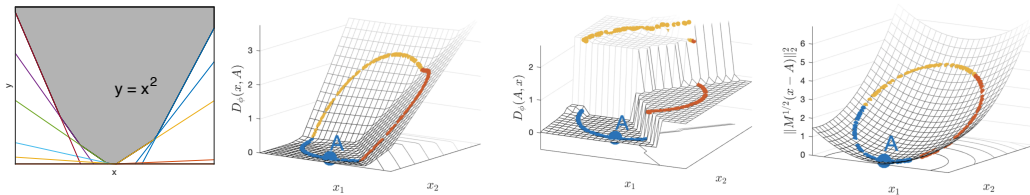

Figure 1: (Left) Approximating a quadratic function via a max-affine function. (Middle-left) Bregman divergence approximation from every 2-d sample point to the specific point $A$ in the data, as $x$ varies around the circle. $x$ to the specific point $A$ in the data (Middle-right) Switches the roles of $x$ and $A$ (recall the BD is asymmetric) (Right) distances from points x to A using a Mahalanobis distance learned via linear metric learning (ITML). When this BD is used to define a Bregman divergence, points within a given class have a small learned divergence, leading to clustering, k-nn, and ranking performance of 98%+ (see experimental results for details).

similar/dissimilar pairs ($B$ and $A$ are similar, $B$ and $C$ are dissimilar) [10]. This supervision may be available when underlying training labels are not directly available, such as from ranking data [20], but can also be obtained directly from class labels in a classification task. In each of these settings, the learned similarity measure can be used downstream as the distance measure in a nearest neighbor algorithm, for similarity-based clustering [3, 19], to perform ranking [23], or other tasks.

Existing metric learning approaches are often divided into two classes, namely linear and non-linear methods. Linear methods learn linear mappings and compute distances (usually Euclidean) in the mapped space [10, 26, 11]; this approach is typically referred to as Mahalanobis metric learning. These methods generally yield simple convex optimization problems, can be analyzed theoretically [4, 8], and are applicable in many general scenarios. Non-linear methods, most notably deep metric learning algorithms, can yield superior performance but require a significant amount of data to train and have little to no associated theoretical properties [28, 16]. As Mahlanaobis distances themselves are within the class of Bregman divergences, this paper shows how one can generalize the class of linear methods to encompass a richer class of possible learned divergences, including non-linear divergences, while retaining the strong theoretical guarantees of the linear case.

To highlight our main contributions, we

- Provide an explicit approximation error bound showing that piecewise linear functions can be used to approximate an underlying Bregman divergence with error $\mathcal{O}(K^{-1/d})$

- Discuss a generalization error bound for metric learning in the Bregman setting of $\mathcal{O}_p(m^{-1/2})$, where $m$ is the number of training points; this matches the bound known for the strictly less general Mahalanobis setting [4]

- Empirically validate our approach problems of ranking and clustering, showing that our method tends to outperform a wide range of linear and non-linear metric learning baselines.

Due to space constraints, many additional details and results have been put into the supplementary material; these include proofs of all bounds, discussion of the regression setting, more details on algorithms, and additional experimental results.

## 2 Related work

To our knowledge, the only existing work on approximating a Bregman divergence is [27], but this work does not provide any statistical guarantees. They assume that the underlying convex function is of the form $\phi(\boldsymbol{x}) = \sum_{i=1}^{N} \alpha_i h(\boldsymbol{x}^T \boldsymbol{x}_i)$, $\alpha_i \geq 0$, where $h(\cdot)$ is a pre-specified convex function such as $|z|^d$. Namely, it is a linear superposition of known convex functions $h(\cdot)$ evaluated on all of the training data. In our preliminary experiments, we have found this assumption to be quite restrictive and falls well short of state-of-art accuracy on benchmark datasets. In contrast to their work, we consider a piecewise linear family of convex functions capable of approximating any convex function. Other relevant non-linear methods include the kernelization of linear methods, as discussed in [19]

and [10]; these methods require a particular kernel function and typically do not scale well for large data.

Linear metric learning methods find a linear mapping $\boldsymbol{G}$ of the input data and compute (squared) Euclidean distance in the mapped space. This is equivalent to learning a positive semi-definite matrix $\boldsymbol{M} = \boldsymbol{G}^T \boldsymbol{G}$ where $d_{\boldsymbol{M}}(\boldsymbol{x}_1, \boldsymbol{x}_2) = (\boldsymbol{x}_1 - \boldsymbol{x}_2)^T \boldsymbol{M}(\boldsymbol{x}_1 - \boldsymbol{x}_2) = \|\boldsymbol{G}\boldsymbol{x}_1 - \boldsymbol{G}\boldsymbol{x}_2\|_2^2$. The literature on linear metric learning is quite large and cannot be fully summarized here; see the surveys [19, 5] for an overview of several approaches. One of the prominent approaches in this class is information theoretic metric learning (ITML) [10], which places a LogDet regularizer on $\boldsymbol{M}$ while enforcing similarity/dissimilarity supervisions as hard constraints for the optimization problem. Large-margin nearest neighbor (LMNN) metric learning [26] is another popular Mahalanobis metric learning algorithm tailored for k-nn by using a local neighborhood loss function which encourages similarly labeled data points to be close in each neighborhood while leaving the dissimilar labeled data points away from the local neighborhood. In Schultz and Joachims [24], the authors use pairwise similarity comparisons ($B$ is more similar to $A$ than to $C$) by minimizing a margin loss.

## 3   Problem Formulation and Approach

We now turn to the general problem formulation considered in this paper. Suppose we observe data points $X = [\boldsymbol{x}_1, ..., \boldsymbol{x}_n]$, where each $\boldsymbol{x}_i \in \mathbb{R}^d$. The goal is to learn an appropriate divergence measure for pairs of data points $\boldsymbol{x}_i$ and $\boldsymbol{x}_j$, given appropriate supervision. The class of divergences considered here is Bregman divergences; recall that Bregman divergences are parameterized by a continuously differentiable, strictly convex function $\phi : \Omega \to \mathbb{R}$, where $\Omega$ is a closed convex set. The Bregman divergence associated with $\phi$ is defined as

$$D_\phi(\boldsymbol{x}_i, \boldsymbol{x}_j) = \phi(\boldsymbol{x}_i) - \phi(\boldsymbol{x}_j) - \nabla\phi(\boldsymbol{x}_j)^T(\boldsymbol{x}_i - \boldsymbol{x}_j).$$

Examples include the squared Euclidean distance (when $\phi(\boldsymbol{x}) = \|\boldsymbol{x}\|_2^2$), the Mahalanobis distance, and the KL divergence. Learning a Bregman divergence can be equivalently described as learning the underlying convex function for the divergence. In order to fully specify the learning problem, we must determine both a supervised loss function as well as a method for appropriately parameterizing the convex function to be learned. Below, we describe both of these components.

### 3.1   Loss Functions

We can easily generalize the standard empirical risk minimization framework for metric learning, as discussed in [19], to our more general setting. In particular, suppose we have supervision in the form of $m$ loss functions $\ell_t$; these $\ell_t$ depend on the learned Bregman divergence parameterized by $\phi$ as well as the data points $X$ and some corresponding supervision $y$. We can express a general loss function as

$$\mathcal{L}(\phi) = \sum_{t=1}^{m} \ell_t(D_\phi, X, y) + \lambda r(\phi),$$

where $r$ is a regularizer over the convex function $\phi$, $\lambda$ is a hyperparameter that controls the tradeoff between the loss and the regularizer, and the supervised losses $\ell_t$ are assumed to be a function of the Bregman divergence corresponding to $\phi$. The goal in an empirical risk minimization framework is to find $\phi$ to minimize this loss, i.e., $\min_{\phi \in \mathcal{F}} \mathcal{L}(\phi)$, where $\mathcal{F}$ is the set of convex functions over which we are optimizing.

The above general loss can capture several learning problems. For instance, one can capture a regression setting, e.g., when the loss $\ell_t$ is the squared loss between the true Bregman divergence and the divergence given by the approximation. In the metric learning setting, one can utilize a loss function $\ell_t$ such as a triplet or contrastive loss, as is standard. In our experiments and generalization error analysis, we mainly consider a generalization of the triplet loss, where the loss is $\max(0, \alpha + D_\phi(\boldsymbol{x}_{i_t}, \boldsymbol{x}_{j_t}) - D_\phi(\boldsymbol{x}_{k_t}, \boldsymbol{x}_{l_t}))$ for a tuple $(\boldsymbol{x}_{i_t}, \boldsymbol{x}_{j_t}, \boldsymbol{x}_{k_t}, \boldsymbol{x}_{l_t})$; see Section 3.3 for details.

### 3.2   Convex piecewise linear fitting

Next we must appropriately parameterize $\phi$. We choose to parameterize our Bregman divergences using piecewise linear approximations. Piecewise linear functions are used in many different applications such as global optimization [22], circuit modeling [17, 14] and convex regression [6, 2]. There

are many methods for fitting piecewise linear functions including using neural networks [12] and local linear fits on adaptive selected partitions of the data [15]; however, we are interested in formulating a convex optimization problem as done in [21]. We use convex piecewise linear functions of the form $\mathcal{F}_{P,L} \doteq \{h : \Omega \to \mathbb{R} \mid h(\boldsymbol{x}) = \max_{k \in [K]} \boldsymbol{a}_k^T \boldsymbol{x} + b_k , \quad \|\boldsymbol{a}_k\|_1 \leq L\}$, called max-affine functions. In our notation $[K]$ denotes the set $\{1, \ldots, K\}$. See the left plot of Figure 1 for a visualization of using a max-affine function.

We stress that our goal is to approximate Bregman divergences, and as such strict convexity and differentiability are not required of the class of approximators when approximating an arbitrary Bregman divergence. Indeed, it is standard practice in learning theory to approximate a class of functions within a more tractable class. In particular, the use of piecewise linear functions has precedence in function approximation, and has been used extensively for approximating convex functions (e.g. [1]).

Conventional numerical schemes seek to approximate a function as a linear superposition of fixed basis functions (eg. Bernstein polynomials). Our method could be directly extended to such basis functions and can be kernelized as well. Still, piecewise linear functions offer a benefit over linear superpositions. The max operator acts as a ridge function resulting in significantly richer non-linear approximations.

In the next section we will discuss how to formulate optimization over $\mathcal{F}_{P,L}$ in order to solve the loss function described earlier. In particular, the following lemma will allow us to express appropriate optimization problems using linear inequality constraints:

**Lemma 1.** *[6] There exists a convex function $\phi : \mathbb{R}^d \to \mathbb{R}$, that takes values*

$$\phi(\boldsymbol{x}_i) = z_i \tag{1}$$

*if and only if there exists $\boldsymbol{a}_1, \ldots, \boldsymbol{a}_n \in \mathbb{R}^d$ such that*

$$z_i - z_j \geq \boldsymbol{a}_j^T (\boldsymbol{x}_i - \boldsymbol{x}_j), \quad i, j \in [n]. \tag{2}$$

*Proof.* Assuming such $\phi$ exists, take $\boldsymbol{a}_j$ to be any sub-gradient of $\phi(\boldsymbol{x}_j)$ then (2) holds by convexity. Conversely, assuming (2) holds, define $\phi$ as

$$\phi(\boldsymbol{x}) = \max_{i \in [n]} \boldsymbol{a}_i^T (\boldsymbol{x} - \boldsymbol{x}_i) + z_i. \tag{3}$$

$\phi$ is convex due to the proposed function being a max of linear functions. $\phi(\boldsymbol{x}_i) = b_i$ using (2). $\quad\square$

As a direct consequence of Lemma 1, one can see that a Bregman divergence can take values

$$D_\phi(\boldsymbol{x}_i, \boldsymbol{x}_j) = z_i - z_j - \boldsymbol{a}_j^T (\boldsymbol{x}_i - \boldsymbol{x}_j), \tag{4}$$

if and only if conditions in (2) hold.

A key question is whether piecewise linear functions can be used to approximate Bregman divergences well enough. An existing result in [1] says that for any $L$-Lipschitz convex function $\phi$ there exists a piecewise linear function $h \in \mathcal{F}_{P,L}$ such that $\|\phi - h\|_\infty \leq 36LRK^{-\frac{2}{d}}$, where $K$ is the number of hyperplanes and $R$ is the radius of the input space. However, this existing result is not directly applicable to us since a Bregman divergence utilizes the gradient $\nabla\phi$ of the convex function. As a result, in section 3.4, we bound the gradient error $\|\nabla\phi - \nabla h\|_\infty$ of such approximators. This in turn allows us to prove a result demonstrating that we can approximate Bregman divergences with arbitrary accuracy under some regularity conditions.

### 3.3 Metric Learning Algorithm

We now briefly discuss algorithms for solving the underlying loss functions described in the previous section. A standard metric learning scenario considers the case where the supervision is given as relative comparisons between objects.

Suppose we observe $S_m = \{(\boldsymbol{x}_{i_t}, \boldsymbol{x}_{j_t}, \boldsymbol{x}_{k_t}, \boldsymbol{x}_{l_t}) \mid t \in [m]\}$, where $D(\boldsymbol{x}_{i_t}, \boldsymbol{x}_{j_t}) \leq D(\boldsymbol{x}_{k_t}, \boldsymbol{x}_{l_t})$ for some unknown similarity function and $(i_t, j_t, k_t, l_t)$ are indices of objects in a countable set $\mathcal{U}$ (e.g.

set of people in a social network). To model the underlying similarity function $D$, we propose a Bregman divergence of the form:

$$\hat{D}(\boldsymbol{x}_i, \boldsymbol{x}_j) \triangleq \hat{\phi}(\boldsymbol{x}_i) - \hat{\phi}(\boldsymbol{x}_j) - \nabla_* \hat{\phi}(\boldsymbol{x}_j), \tag{5}$$

where $\hat{\phi}(\boldsymbol{x}) \triangleq \max_{i \in \mathcal{U}_m} \boldsymbol{a}_i^T(\boldsymbol{x} - \boldsymbol{x}_i) + z_i$ , $\nabla_*$ is the biggest sub-gradient, $\mathcal{U}_m$ is the set of all observed objects indices $\mathcal{U}_m \triangleq \cup_{t=1}^m \{i_t, j_t, k_t, l_t\}$ and $\boldsymbol{a}_i$'s and $z_i$'s are the solution to the following linear program:

$$\min_{z_i, \boldsymbol{a}_i, L} \sum_{t=1}^m \max(\zeta_t, 0) + \lambda L$$

$$\text{s.t.} \begin{cases} z_{i_t} - z_{j_t} - \boldsymbol{a}_{j_t}^T(\boldsymbol{x}_{i_t} - \boldsymbol{x}_{j_t}) + z_{l_t} - z_{k_t} + \boldsymbol{a}_{l_t}^T(\boldsymbol{x}_{k_t} - \boldsymbol{x}_{l_t}) \leq \zeta_t - 1 & t \in [m] \\ z_i - z_j \geq \boldsymbol{a}_j^T(\boldsymbol{x}_i - \boldsymbol{x}_j) & i, j \in \mathcal{U}_m \\ \|\boldsymbol{a}_i\|_1 \leq L & i \in \mathcal{U}_m \end{cases}$$

We refer to the solution of this optimization problem as PBDL (piecewise Bregman divergence learning). Note that one may consider other forms of supervision, such as pairwise similarity constraints, and these can be handled in an analogous manner. Also, the above algorithm is presented for readability for the case where $K = n$; the case where $K < n$ is discussed in the supplementary material.

In order to scale our method to large datasets, there are several possible approaches. One could employ ADMM to the above LP, which can be implemented in a distributed fashion or on GPUs.

## 3.4 Analysis

Now we present an analysis of our approach. Due to space considerations, proofs appear in the supplementary material. Briefly, our results: i) show that a Bregman divergence parameterized by a piecewise linear convex function can approximate an arbitrary Bregman divergence with error $\mathcal{O}(K^{-\frac{1}{d}})$, where $K$ is the number of affine functions; ii) bound the Rademacher complexity of the class of Bregman divergences parameterized by piecewise linear generating functions; iii) provide a generalization for Bregman metric learning that shows that the generalization error gap grows as $\mathcal{O}_p(m^{-\frac{1}{2}})$, where $m$ is the number of training points.

In the supplementary material, we further provide additional generalization guarantees for learning Bregman divergences in the regression setting. In particular, it is worth noting that, in the regression setting, we provide a generalization bound of $\mathcal{O}_p(m^{-1/(d+2)})$, which is comparable to the lower-bound for convex regression $\mathcal{O}_p(m^{-4/(d+4)})$.

**Approximation Guarantees.** First we would like to bound how well one can approximate an arbitrary Bregman divergence when using a piecewise linear convex function. Besides providing a quantitative justification for using such generating functions, this result is also used for later generalization bounds.

**Theorem 1.** *For any convex $\phi : \Omega \to \mathbb{R}$, which:*
*1) is defined on the $\infty$-norm ball, i.e:*

$$\mathcal{B}(R) = \{\boldsymbol{x} \in \mathbb{R}^d, \|\boldsymbol{x}\|_\infty \leq R\} \subset \Omega$$

*2) is $\beta$-smooth, i.e:*

$$\|\nabla \phi(\boldsymbol{x}) - \nabla \phi(\boldsymbol{y})\|_1 \leq \beta \|\boldsymbol{x} - \boldsymbol{y}\|_\infty.$$

*There exists a max-affine function $h$ with $K$ hyper-planes such that:*
*1) it uniformly approximates $\phi$:*

$$\sup_{\boldsymbol{x} \in \mathcal{B}(R)} |\phi(\boldsymbol{x}) - h(\boldsymbol{x})| \leq 4\beta R^2 K^{-2/d}. \tag{6}$$

*2) Any of its sub-gradients $\nabla h(\boldsymbol{x}) \in \partial h(\boldsymbol{x})$ away from boundaries of the norm ball, uniformly approximates $\nabla \phi(\boldsymbol{x})$.*

$$\sup_{\boldsymbol{x} \in \mathcal{B}(R-\epsilon)} \|\nabla \phi(\boldsymbol{x}) - \nabla h(\boldsymbol{x})\|_1 \leq 16\beta R K^{-1/d}, \tag{7}$$

*3) The Bregman divergence parameterized by $h$ away from boundaries of the norm ball, uniformly approximates Bregman divergence parameterized by $\phi$*

$$\sup_{\boldsymbol{x},\boldsymbol{x}'\in\mathcal{B}(R-\epsilon)}|D_\phi(\boldsymbol{x},\boldsymbol{x}') - D_h(\boldsymbol{x},\boldsymbol{x}')| \leq 36\beta R^2 K^{-1/d}, \tag{8}$$

$$\epsilon \leq 8RK^{-1/d}.$$

**Rademacher Complexity.** Another result we require for proving generalization error is the Rademacher complexity of the class of Bregman divergences using our choice of generating functions. We have the following result:

**Lemma 2.** *The Radamacher complexity of Bregman divergences parameterized by max-affine functions, $R_m(\mathcal{D}_{P,L})$, is bounded by $R_m(\mathcal{D}_{P,L}) \leq 4KLR\sqrt{2\ln(2d+2)/m}$.*

**Generalization Error.** Finally, we consider the case of classification error when learning a Bregman divergence under relative similarity constraints. Our result bounds the loss on unseen data based on the loss on the training data. We require that the training data be drawn iid. Note that while there are known methods to relax these assumptions, as shown for Mahalanobis metric learning in [4], we assume here for simplicity that data is drawn iid.[1] In particular, we assume that each instance is a quintuple, consisting of two pairs $(\boldsymbol{x}_{i_t}, \boldsymbol{x}_{j_t}, \boldsymbol{x}_{k_t}, \boldsymbol{x}_{l_t})$ drawn iid from some distribution $\mu$ over $\mathcal{X}^4$.

**Theorem 2.** *Consider $S_m = \{(\boldsymbol{x}_{i_t}, \boldsymbol{x}_{j_t}, \boldsymbol{x}_{k_t}, \boldsymbol{x}_{l_t}), t \in [m]\} \sim \mu^m$, where $D(\boldsymbol{x}_{i_t}, \boldsymbol{x}_{j_t}) \leq D(\boldsymbol{x}_{k_t}, \boldsymbol{x}_{l_t})$. Set $R = \max_i \|\boldsymbol{x}_i\|_\infty$. The generalization error of the learned divergence in (1) when using $K$ hyper-planes satisfies*

$$\mathbb{E}\big[\mathbb{1}[\hat{D}(\boldsymbol{x}_{i_t}, \boldsymbol{x}_{j_t}) \geq \hat{D}(\boldsymbol{x}_{k_t}, \boldsymbol{x}_{l_t})]\big] \leq \frac{1}{m}\sum_{t=1}^m \max\big(0, 1 + \hat{D}(\boldsymbol{x}_{i_t}, \boldsymbol{x}_{j_t}) - \hat{D}(\boldsymbol{x}_{k_t}, \boldsymbol{x}_{l_t})\big)$$
$$+ 32KLR\sqrt{2\ln(2d+2)}/\sqrt{m}$$
$$+ \sqrt{4\ln(4\log_2 L) + \ln(1/\delta)}/\sqrt{m}$$

*with probability at least $1-\delta$ for receiving the data $S_m$.*

See the supplementary material for a proof.

**Discussion of Theorem 2:** Not that $n$ stands for number of unique points in all comparisons, where $m$ stands for number of comparisons, i.e: $n = \#\mathcal{U}_m$, so $n$ will increase with $m$.

**case 1: (K<n)** We discuss details of the algorithm about the case where $K < n$ in appendix A6; this is another approach we have used in our experiments which yielded similar results. Using standard cross-validation to select K is the simplest and most effective way to select a value of K, and also ensures that the theoretical bounds are applicable. Also its almost obvious that doing further cross validation to choose $K$ would result in improvement over the choice of $K = n$. However we liked to use $K = n$ in the reported experiments as it results in a faster algorithm and reduces the time needed for cross validation.

**case 2: (K=n)** For the theoretical bound to hold we need $n << m$. This could be true in the example of the social network if we extract some kind of similarity information between people. Regardless of this we found acceptable results in our experiments with this setting.

**i.i.d setup:** The i.i.d setup while training is enforced by randomly choosing the similarity comparisons from the fixed classification data-set $\{\boldsymbol{x}_i, y_i\}_{i=1}^n$. This makes sense in a practical sense too as having or computing relative comparisons between all triplets would make $m = O(n^3)$ which is impractical when $n$ is large. However we test the divergence in different tasks (i.e. ranking and clustering).

## 4 Experiments

Due to space constraints, we focus mainly on comparisons to Mahalanobis metric learning methods and their variants for the problems of clustering and ranking. In the supplementary, we include

Table 1: Learning Bregman divergences (PDBL) compared to existing linear and non-linear metric learning approaches on standard UCI benchmarks. PDBL performs first or second among these benchmarks in 14 of 16 comparisons, outperforming all of the other methods. Note that the top two results for each setting are indicated in bold.

| Data-set | Algorithm | Clustering | | Ranking | |
|---|---|---|---|---|---|
| | | Rand-Ind % | Purity % | AUC % | Ave-P % |
| Iris | PBDL | $\mathbf{94.5 \pm 0.8}$ | $\mathbf{95.6 \pm 0.7}$ | $\mathbf{96.5 \pm 0.4}$ | $\mathbf{93.5 \pm 0.7}$ |
| | ITML [10] | $\mathbf{96.4 \pm 0.8}$ | $\mathbf{97.0 \pm 0.7}$ | $\mathbf{97.5 \pm 0.3}$ | $\mathbf{95.3 \pm 0.5}$ |
| | LMNN [26] | $90.0 \pm 1.3$ | $91.0 \pm 1.3$ | $94.3 \pm 0.6$ | $89.9 \pm 0.8$ |
| | GB-LMNN [18] | $88.7 \pm 1.5$ | $89.9 \pm 1.5$ | $94.0 \pm 0.6$ | $89.7 \pm 0.8$ |
| | GMML [29] | $93.8 \pm 0.9$ | $94.5 \pm 0.9$ | $95.7 \pm 0.4$ | $92.0 \pm 0.6$ |
| | Kernel NCA [11] | $89.9 \pm 1.3$ | $90.3 \pm 1.1$ | $93.4 \pm 0.6$ | $88.3 \pm 0.9$ |
| | MLR-AUC [23] | $79.7 \pm 2.6$ | $80.1 \pm 2.7$ | $84.2 \pm 2.4$ | $76.2 \pm 2.6$ |
| | Euclidean | $87.8 \pm 0.8$ | $89.2 \pm 0.8$ | $93.5 \pm 0.3$ | $88.8 \pm 0.5$ |
| Balance Scale | PBDL | $\mathbf{84.4 \pm 0.7}$ | $\mathbf{87.8 \pm 0.5}$ | $\mathbf{86.0 \pm 0.4}$ | $\mathbf{82.9 \pm 0.5}$ |
| | ITML | $68.9 \pm 0.9$ | $77.5 \pm 0.7$ | $\mathbf{80.1 \pm 0.7}$ | $\mathbf{74.3 \pm 0.8}$ |
| | LMNN | $69.5 \pm 1.8$ | $77.0 \pm 1.7$ | $75.9 \pm 1.3$ | $70.0 \pm 1.2$ |
| | GB-LMNN | $71.4 \pm 1.5$ | $79.7 \pm 1.4$ | $78.1 \pm 1.1$ | $72.2 \pm 1.0$ |
| | GMML | $\mathbf{72.9 \pm 0.8}$ | $\mathbf{80.2 \pm 0.8}$ | $79.0 \pm 0.4$ | $72.8 \pm 0.5$ |
| | NCA | $58.9 \pm 0.5$ | $65.9 \pm 0.8$ | $66.7 \pm 0.2$ | $61.7 \pm 0.3$ |
| | Kernel NCA | $65.3 \pm 1.5$ | $73.0 \pm 1.6$ | $68.7 \pm 1.8$ | $63.7 \pm 1.9$ |
| | MLR-AUC | $48.0 \pm 2.6$ | $53.6 \pm 2.9$ | $56.9 \pm 3.3$ | $55.8 \pm 3.1$ |
| | Euclidean | $59.3 \pm 0.6$ | $66.5 \pm 0.9$ | $67.9 \pm 0.3$ | $66.2 \pm 0.4$ |
| Wine | PBDL | $\mathbf{83.7 \pm 2.9}$ | $\mathbf{85.0 \pm 3.2}$ | $\mathbf{91.0 \pm 0.9}$ | $\mathbf{86.7 \pm 1.2}$ |
| | ITML | $82.8 \pm 2.6$ | $82.5 \pm 3.1$ | $89.1 \pm 1.1$ | $84.6 \pm 1.4$ |
| | LMNN | $70.0 \pm 0.8$ | $68.8 \pm 1.2$ | $82.4 \pm 0.8$ | $76.2 \pm 1.1$ |
| | GB-LMNN | $70.6 \pm 0.9$ | $69.3 \pm 1.4$ | $83.7 \pm 0.1$ | $78.5 \pm 1.3$ |
| | GMML | $\mathbf{83.2 \pm 2.9}$ | $\mathbf{81.0 \pm 3.2}$ | $\mathbf{91.0 \pm 0.7}$ | $\mathbf{88.5 \pm 0.7}$ |
| | Kernel NCA | $70.4 \pm 1.3$ | $71.3 \pm 1.4$ | $75.1 \pm 0.9$ | $67.7 \pm 1.1$ |
| | MLR-AUC | $33.1 \pm 1.0$ | $40.4 \pm 1.4$ | $52.5 \pm 1.5$ | $52.4 \pm 1.5$ |
| | Euclidean | $71.2 \pm 0.7$ | $70.6 \pm 0.8$ | $77.7 \pm 0.7$ | $66.1 \pm 0.8$ |
| Transfusion | PBDL | $57.9 \pm 1.2$ | $75.9 \pm 0.7$ | $\mathbf{54.9 \pm 0.4}$ | $\mathbf{68.2 \pm 0.6}$ |
| | ITML | $60.2 \pm 1.0$ | $75.8 \pm 0.7$ | $54.2 \pm 0.4$ | $66.4 \pm 0.7$ |
| | LMNN | $59.4 \pm 1.3$ | $76.3 \pm 0.6$ | $54.0 \pm 0.5$ | $67.1 \pm 0.7$ |
| | GB-LMNN | $58.9 \pm 1.2$ | $76.3 \pm 0.6$ | $\mathbf{54.8 \pm 0.6}$ | $67.2 \pm 0.7$ |
| | GMML | $59.3 \pm 1.3$ | $\mathbf{76.6 \pm 0.7}$ | $54.0 \pm 0.5$ | $\mathbf{67.5 \pm 0.7}$ |
| | Kernel NCA | $\mathbf{63.7 \pm 0.7}$ | $76.2 \pm 0.7$ | $52.2 \pm 0.8$ | $65.7 \pm 0.8$ |
| | MLR-AUC | $\mathbf{62.6 \pm 1.8}$ | $74.9 \pm 2.2$ | $42.8 \pm 1.3$ | $60.9 \pm 1.8$ |
| | Euclidean | $60.6 \pm 0.6$ | $\mathbf{76.4 \pm 0.4}$ | $54.2 \pm 0.3$ | $67.0 \pm 0.5$ |

several additional empirical results: i) additional data sets, ii) comparisons on k-nearest neighbor classification performance with the learned metrics, iii) results for the regression learning setting.

In the following, all results are represented using $95\%$ confidence intervals, computed using 100 runs. Our optimization problems are solved using Gurobi solvers [13]. We compared against both linear and kernelized Mahalanobis metric learning methods, trying to include as many popular linear and non-linear approaches within our space limitations. In particular, we compared to 8 baselines: information-theoretic metric learning (ITML) [10], large-margin nearest neighbors (LMNN) [26], gradient-boosted LMNN (GB-LMNN)[18], geometric mean metric learning (GMML)[29], neighbourhood components analysis (NCA) [11], kernel NCA, metric learning to rank (MLR)[23], and a baseline Euclidean distance. Note that we also compared to the Bregman divergence learning algorithm of Wu et al. [27] but found its performance in general to be much worse than the other metric learning methods; see the discussion below. Code for all experiments is available on our github page[2].

### 4.1 Bregman clustering and similarity ranking from relative similarity comparisons

In this experiment we implement PBDL on four standard UCI classification data sets that have previously been used for metric learning benchmarking. See the supplementary material for additional data sets. We apply the learned divergences to the tasks of semi-supervised clustering and similarity ranking. To learn a Bregman divergence we use a cross-validation scheme with 3 folds. From two folds we learn the Bregman divergence or Mahalanobis distance and then test it for the specified task on the other fold. All results are summarized in Table 1.

**Data**: The pairwise inequalities are generated by choosing two random samples $\boldsymbol{x}_1, \boldsymbol{x}_2$ from a random class and another sample $\boldsymbol{x}_3$ from a different class. We provided the supervision $D(\boldsymbol{x}_1, \boldsymbol{x}_2) \leq D(\boldsymbol{x}_1, \boldsymbol{x}_3)$. The number of inequalities provided was 2000 for each case.

**Divergence learning details**: The $\lambda$ in our algorithm (PBDL) were both chosen by 3-fold cross validation on training data on a grid $10^{-8:1:4}$. For implementing ITML we used the original code and the hyper-parameters were optimized by a similar cross-validation using their tuner for each different task. We used the code provided in Matlab statistical and machine learning toolbox for a diagonal version of NCA. We Kernelized NCA by the kernel trick where we chose the kernel by cross validation to be either RBF or polynomial kernel with bandwidth chosen from $2^{0:5}$. For GB-LMNN we used their provided code. We performed hyper-parameter tuning for GMML as described in their paper. For MLR-AUC we used their code and guidelines for hyper-parameter optimization.

For the clustering task, it was shown in [3] that one can do clustering similarly to k-means for any Bregman divergence. We use the learned Bregman divergence to do Bregman clustering and measure the performance under *Rand-Index* and *Purity*. For the ranking task, for each test data point $\boldsymbol{x}$ we rank all other test data points according to their Bergman divergence. The ground truth ranking is one where for any data point $\boldsymbol{x}$ all similarly labeled data points are ranked earlier than any data from other classes. We evaluate the performance by computing *average-precision* (Ave-P) and *Area under ROC curve* (AUC) on test data as in [23].

Also [9] considers extensions of the framework considered in this paper to the deep setting; they show that one can achieve state-of-the-art results for image classification, and the Bregman learning framework outperforms popular deep metric learning baselines on several tasks and data sets. Thus, we may view a contribution of our paper as building a theoretical framework that already has shown impact in deep settings.

### 4.2 Discussion and Observations

On the benchmark datasets examined, our method yields the best or second-best results on 14 of the 16 comparisons (4 datasets by 4 measures per dataset); the next best method (GMML) yields best or second-best results on 8 comparisons. This suggests that Bregman divergences are competitive for downstream clustering and ranking tasks.

We spent quite a bit of time working with the method of [27], which learns a convex function of the form $\phi(\boldsymbol{x}) = \sum_{i=1}^{N} \alpha_i h(\boldsymbol{x}^T \boldsymbol{x}_i)$, $\alpha_i \geq 0$, but we found that the algorithm did not produce sensible results. More precisely: the solution oscillated between two solutions, one that classifies every-pair as similar and the other one classifying every pair as dis-similar. Also, when tuning the learning rate carefully, the oscillations converged to $\alpha_i = 0, \forall i$. We think the problem is not only with the algorithm but with the formulation as well. The authors of [27] recommend two different kernels: $\phi(\boldsymbol{x}) = \sum_i \exp(\boldsymbol{x}^T \boldsymbol{x}_i)$ and $\phi(\boldsymbol{x}) = \sum_i \alpha_i (\boldsymbol{x}^T \boldsymbol{x}_i)^2$. Adding $n$ exponential functions results in a very large and unstable function. The same holds for adding quadratic functions. In our formulation we are taking max of $n$ linear functions so at each instance only one linear functions is active.

### 4.3 Conclusions

We developed a framework for learning arbitrary Bregman divergences by using max-affine generating functions. We precisely bounded the approximation error of such functions as well as provided generalization guarantees in the regression and relative similarity setting.

## Broader Impacts

The metric learning problem is a fundamental problem in machine learning, attracting considerable research and applications. These applications include (but are certainly not limited to) face verification, image retrieval, human activity recognition, program debugging, music analysis, and microarray data analysis (see [19] for a discussion of each of these applications, along with relevant references). Fundamental work in this problem will help to improve results in these applications as well as lead to further impact in new domains. Moreover, a solid theoretical understanding of the algorithms and methods of metric learning can lead to improvements in combating learning bias for these applications and reduce unnecessary errors in several systems.

## 5 Acknowledgment

This research was supported by NSF CAREER Award 1559558, CCF-2007350 (VS), CCF-2022446 (VS), CCF-1955981 (VS) and the Data Science Faculty Fellowship from the Rafik B. Hariri Institute. We would like to thank Gábor Balázs for helpful comments and suggestions. We thank Natalia Frumkin for helping us with experiments. We thank Arian Houshmand for providing suggestions that led to speeding up our linear programming.

## Footnotes

[1]In many cases this is justified. For instance, in estimating quality scores for items, one often has data corresponding to item-item comparisons [25]; for each item, the learner also observes contextual information. The feedback, $y_t$ depends only on the pair $(\boldsymbol{x}_{i_t}, \boldsymbol{x}_{j_t})$, and as such is independent of other comparisons.

[2]https://github.com/Siahkamari/Learning-to-Approximate-a-Bregman-Divergence.git

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
