[Supplementary Material]

# Learning to Approximate a Bregman Divergence: Supplementary Material

## 1 Appendix

In this appendix, we provide several additional results:

- Background mathematical concepts required for the proofs (A1)
- Proof of the approximation bound (A2)
- Proof of the Rademacher complexity bound (A3)
- Proof of the metric learning generalization error bound (A4)
- Discussion of the regression setting, including generalization error (A5)
- Discussion of the case where $K < n$ (A6)
- Some additional details omitted from the discussion of the algorithms (A7)
- Additional experimental results, including results on regression and classification (A8)

### A1 Covering number

This is a brief overview of covering numbers from [1]. Let $(\Omega, \|\cdot\|)$ be a metric space and $\Omega \subset \mathbb{U}$. For any $\epsilon > 0$, $\mathbb{X}_\epsilon \subset \mathbb{U}$ is an $\epsilon$-covering of $\Omega$ if:

$$\min_{\hat{\boldsymbol{x}} \in \mathbb{X}_\epsilon} \|\boldsymbol{x} - \hat{\boldsymbol{x}}\| \leq \epsilon \quad \forall \boldsymbol{x} \in \Omega.$$

The covering number $\mathcal{N}(\Omega, \epsilon, \|\cdot\|)$ is defined as the minimum cardinallity of an $\epsilon$-covering of $\Omega$. By volumetric arguments, the covering number of the norm ball of radius $R$ in $d$-dimension $\mathcal{B}(R)$ is bounded as below:

$$\left(\frac{R}{\epsilon}\right)^d \leq \mathcal{N}(\mathcal{B}(R), \epsilon, \|\cdot\|) \leq \left(\frac{2R}{\epsilon} + 1\right)^d.$$

In this paper we only consider the $\|\cdot\|_\infty$ on the input space. We construct a covering set by dividing the space into hyper-cubes of side length $2\epsilon$ as depicted in Figure 1. This construction provides us a covering set of size $\mathcal{N}(\mathcal{B}(R), \epsilon, \|\cdot\|_\infty) \leq \lceil R/\epsilon \rceil^d$.

### A2 Approximation Guarantees

Before proceeding with the proof we state a useful lemma.

**Lemma 1.** *[6] For any two vectors $\boldsymbol{r}_1, \boldsymbol{r}_2 \in \mathbb{R}^d$,*

$$\sup_{\|\boldsymbol{u}\|_\infty \leq \rho} \langle \boldsymbol{r}_1 - \boldsymbol{r}_2, \boldsymbol{u} \rangle \leq \delta \iff \|\boldsymbol{r}_1 - \boldsymbol{r}_2\|_1 \leq \delta/\rho.$$

Figure 1: Sketch of a 2-dimensional hyper-cube of radius $R$, covered by $\infty$-norm balls of radius $\epsilon$.

**Proof of Theorem 1.**

Let $\mathbb{X}_\epsilon = \{\hat{\boldsymbol{x}}_1, \ldots, \hat{\boldsymbol{x}}_K\}$ be an $\epsilon$-cover for $\mathcal{B}(R)$ as constructed in A1. We have:

$$\epsilon = \frac{R}{\lfloor K^{1/d} \rfloor} \le 2RK^{-1/d}.$$

Let $\hat{\boldsymbol{x}} = \arg\min_{\hat{\boldsymbol{x}}_i \in \mathbb{X}_\epsilon} \|\boldsymbol{x} - \hat{\boldsymbol{x}}_i\|_\infty$. We know $\|\boldsymbol{x} - \hat{\boldsymbol{x}}\|_\infty \le \epsilon$ due to construction of $\mathbb{X}_\epsilon$. Consider the piecewise linear function, $h : \mathbb{R}^d \to \mathbb{R}$, defined as follows:

$$h(\boldsymbol{x}) \triangleq \max_i \phi(\hat{\boldsymbol{x}}_i) + \langle \nabla\phi(\hat{\boldsymbol{x}}_i), \boldsymbol{x} - \hat{\boldsymbol{x}}_i \rangle. \tag{1}$$

We have:

$$
\begin{aligned}
0 \le \phi(\boldsymbol{x}) &- h(\boldsymbol{x}) \\
&\le \phi(\boldsymbol{x}) - \phi(\hat{\boldsymbol{x}}) - \langle \nabla\phi(\hat{\boldsymbol{x}}), \boldsymbol{x} - \hat{\boldsymbol{x}} \rangle \\
&\le \langle \nabla\phi(\boldsymbol{x}) - \nabla\phi(\hat{\boldsymbol{x}}), \boldsymbol{x} - \hat{\boldsymbol{x}} \rangle \\
&\le \|\nabla\phi(\boldsymbol{x}) - \nabla\phi(\hat{\boldsymbol{x}})\|_1 \|\boldsymbol{x} - \hat{\boldsymbol{x}}\|_\infty \\
&\le \beta\|\boldsymbol{x} - \hat{\boldsymbol{x}}\|_\infty^2 \le \beta\epsilon^2 = 4\beta R^2 K^{-2/d}.
\end{aligned}
$$

Therefore (9) in the main paper is shown. For proving (10) consider covering points $\hat{\boldsymbol{x}}_i$ in a $\delta$-ball around $\boldsymbol{x}$.

$$
\begin{aligned}
\langle \nabla&\phi(\boldsymbol{x}) - \nabla h(\boldsymbol{x}), \boldsymbol{x} - \hat{\boldsymbol{x}}_i \rangle \\
&= \langle \nabla\phi(\boldsymbol{x}) - \nabla\phi(\hat{\boldsymbol{x}}_i), \boldsymbol{x} - \hat{\boldsymbol{x}}_i \rangle \\
&\quad + \phi(\hat{\boldsymbol{x}}_i) + \langle \nabla\phi(\hat{\boldsymbol{x}}_i), \boldsymbol{x} - \hat{\boldsymbol{x}}_i \rangle && {}^*\big( < h(\boldsymbol{x}) \big) \\
&\quad - h(\hat{\boldsymbol{x}}_i) - \langle \nabla h(\boldsymbol{x}), \boldsymbol{x} - \hat{\boldsymbol{x}}_i \rangle && {}^{**}\big( < -h(\boldsymbol{x}) \big) \\
&\le \|\nabla\phi(\boldsymbol{x}) - \nabla\phi(\hat{\boldsymbol{x}}_i)\|_1 \|\boldsymbol{x} - \hat{\boldsymbol{x}}_i\|_\infty \\
&\le \beta\|\boldsymbol{x} - \hat{\boldsymbol{x}}_i\|_\infty^2 \le \beta\delta^2. \tag{2}
\end{aligned}
$$

$*$ is true due to the way $h(x)$ is defined in (1). $**$ is true due to convexity. By a convex combination of inequalities in (2) we get:

$$\langle \nabla\phi(\boldsymbol{x}) - \nabla h(\boldsymbol{x}), \boldsymbol{x} - \sum_i \alpha_i \hat{\boldsymbol{x}}_i \rangle \le \beta\delta^2. \tag{3}$$

Next we will prove $\boldsymbol{x} - \sum_i \alpha_i \hat{\boldsymbol{x}}_i$ can represent any vector $\boldsymbol{r}$ of size $\|\boldsymbol{r}\|_\infty \le \delta - 2\epsilon$. From there by using Lemma 1 and choosing $\delta = 4\epsilon$ we'll get

$$\|\nabla\phi(\boldsymbol{x}) - \nabla h(\boldsymbol{x})\|_1 \le \beta\delta^2/(\delta - 2\epsilon) \le 16\beta R K^{-1/d}.$$

35 If $\delta \geq 2\epsilon$ and $\boldsymbol{x}$ is no closer than $\delta$ to the boundaries of $\mathcal{B}(R)$, we can consider hyper-cubes $\mathcal{B}(\epsilon)$
36 fitted to each corner of $\mathcal{B}(\delta)$ centered at $\boldsymbol{x}$ as in the 2-dimensional case depicted by Figure 2. There
37 has to be covering points in each of these $\epsilon$-hyper-cubes, otherwise their center is further away from
38 all covering points by more than $\epsilon$. As depicted by Figure 2 the convex hull of such covering points
39 includes a $(\delta - 2\epsilon)$-hyper-cube centered at $\boldsymbol{x}$. Therefore any vector of size $(\delta - 2\epsilon)$ can be represented
by $(\boldsymbol{x} - \sum_i \alpha_i \hat{\boldsymbol{x}}_i)$.

Figure 2: The 2-dimensional sketch of the input space $\mathcal{B}(R)$ along with $\mathcal{B}(\delta)$ centered at $\boldsymbol{x}$. Four dashed vectors represent $\boldsymbol{x} - \hat{\boldsymbol{x}}_k$. Using a convex combination of these vectors we can represent any vector $\boldsymbol{r}$ (solid vector) of size $\|\boldsymbol{r}\|_\infty = \delta - 2\epsilon$.

41 The proof of (11) in the main paper is done by combining the approximation error of the gradient and
42 the convex function as follows:

$$
\begin{aligned}
D_\phi(\boldsymbol{x}, \boldsymbol{x}') - D_h(\boldsymbol{x}, \boldsymbol{x}') &= \phi(\boldsymbol{x}) - \phi(\boldsymbol{x}') - \langle \nabla\phi(\boldsymbol{x}'), \boldsymbol{x} - \boldsymbol{x}' \rangle \\
&\quad - h(\boldsymbol{x}) + h(\boldsymbol{x}') + \langle \nabla h(\boldsymbol{x}'), \boldsymbol{x} - \boldsymbol{x}' \rangle \\
&\leq \phi(\boldsymbol{x}) - h(\boldsymbol{x}) \\
&\quad + \langle \nabla h(\boldsymbol{x}') - \nabla\phi(\boldsymbol{x}'), \boldsymbol{x} - \boldsymbol{x}' \rangle \\
&\leq |\phi(\boldsymbol{x}) - h(\boldsymbol{x})| \\
&\quad + \|\nabla\phi(\boldsymbol{x}) - \nabla h(\boldsymbol{x})\|_1 \|\boldsymbol{x} - \boldsymbol{x}'\|_\infty \\
&\leq 36\beta R^2 K^{-1/d}.
\end{aligned}
$$

43 The other side of the inequality can be shown similarly.

## A3 Rademacher complexity of piecewise linear Bregman divergences

45 The Rademacher complexity $R_m(\mathcal{F})$ of a function class $\mathcal{F}$ is defined as the expected maximum
46 correlation of a function class with binary noise. Bounding the Radamacher complexity of a function
47 class provides us with a measure of how complex the class is. This measure is used in computing
48 probably approximately correct (PAC) bounds for learning tasks such as classification, regression,
49 and ranking. Let

$$
\mathcal{F}_{P,L} \triangleq \{h : \mathbb{R}^d \to \mathbb{R} \,|\, h(\boldsymbol{x}) = \max_{k \in [K]} \boldsymbol{a}_k^T \boldsymbol{x} + b_k, \|\boldsymbol{a}_k\|_1 \leq L\}
$$

50 be the class of $L$-Lipschitz max-affine functions. Also let

$$
\mathcal{D}_{P,L} \triangleq \{h(\boldsymbol{x}) - h(\boldsymbol{x}') - \nabla h(\boldsymbol{x}')^T(\boldsymbol{x} - \boldsymbol{x}') \,|\, h \in \mathcal{F}_{P,L}\}
$$

51 be the class of Bregman divergences parameterized by a max-affine functions.

<sup>53</sup> **Lemma 2.** *The Radamacher complexity of Bregman divergences parameterized by a max-affine*
<sup>54</sup> *function $R_m(\mathcal{D}_{P,L}) \leq 4KLR\sqrt{(2\ln(2d+2))/m}$.*

<sup>55</sup> Proof. Define: $p(\boldsymbol{x}) \triangleq \arg\max_k \boldsymbol{a}_k^T \boldsymbol{x} + b_k$

$$\mathcal{D}_{P,L} = \{h(\boldsymbol{x}) - h(\boldsymbol{x}') - \nabla h(\boldsymbol{x}')^T(\boldsymbol{x} - \boldsymbol{x}') \mid h \in \mathcal{F}_{P,L}\}$$
$$= \{\boldsymbol{a}_{p(\boldsymbol{x})}^T \boldsymbol{x} + b_{p(\boldsymbol{x})} - \boldsymbol{a}_{p(\boldsymbol{x}')}^T \boldsymbol{x} - b_{p(\boldsymbol{x}')} \mid \|\boldsymbol{a}_i\|_1 \leq L\}$$
$$= \{\boldsymbol{a}_{p(\boldsymbol{x})}^T \boldsymbol{x} + c_{p(\boldsymbol{x})} - \boldsymbol{a}_{p(\boldsymbol{x}')}^T \boldsymbol{x} - c_{p(\boldsymbol{x}')}$$
$$\mid \|\boldsymbol{a}_i\|_1 \leq L, c_i = b_i - b_{p(0)} + LR\}.$$

<sup>56</sup> Note that $|c_i| \leq LR$:
$$-c_i = b_{p(0)} - b_i - LR = \max_k b_k - b_i - LR \geq -LR.$$

<sup>57</sup> For the other side, consider $\boldsymbol{x}$ such that $h(\boldsymbol{x}) = \boldsymbol{a}_i^T \boldsymbol{x} + b_i$. If no such $\boldsymbol{x}$ exists, we can discard the $i^{th}$
<sup>58</sup> hyper-plane. Therefore:

$$-c_i = b_{p(0)} - b_i - LR = \max_k b_k - b_i - LR$$
$$= h(0) - h(\boldsymbol{x}) + \boldsymbol{a}_i^T \boldsymbol{x} - LR$$
$$\leq L\|0 - \boldsymbol{x}\|_\infty + \|\boldsymbol{a}_i\|_1 \|\boldsymbol{x}\|_\infty - LR \leq LR.$$

<sup>59</sup> Now we are ready to compute the Radamacher complexity:

$$R_m(\mathcal{D}_{P,L}) = \frac{1}{m}\mathbb{E}_\sigma \sup \sum_{i=1}^{m} \sigma_i D_h(\boldsymbol{x}_i, \boldsymbol{x}_i')$$

$$= \frac{1}{m}\mathbb{E}_\sigma \sup_{\substack{\forall k\ \|\boldsymbol{a}_k\|_1 \leq L \\ \forall k\ \|c_k\|_1 \leq LR}} \sum_{i=1}^{m} \sigma_i(\boldsymbol{a}_{p(\boldsymbol{x}_i)}^T \boldsymbol{x}_i + c_{p(\boldsymbol{x}_i)}$$
$$- \boldsymbol{a}_{p(\boldsymbol{x}_i')}^T \boldsymbol{x}_i - c_{p(\boldsymbol{x}_i')})$$

$$\leq \frac{2}{m}\mathbb{E}_\sigma \sup_{\substack{\forall k\ \|\boldsymbol{a}_k\|_1 \leq L \\ \forall k\ \|c_k\|_1 \leq LR}} \sum_{i=1}^{m}\sum_{k=1}^{K} |\sigma_i(\boldsymbol{a}_k^T \boldsymbol{x}_i + c_k)|$$

$$= \frac{2K}{m}\mathbb{E}_\sigma \sup_{\substack{\|\boldsymbol{a}_1\|_1 \leq L \\ \|c_1\|_1 \leq LR}} \sum_{i=1}^{m} \left|\sigma_i \begin{bmatrix} c_1/R \\ \boldsymbol{a}_1 \end{bmatrix}^T \begin{bmatrix} R \\ \boldsymbol{x}_i \end{bmatrix}\right|.$$

<sup>60</sup> The last expression is $2K$ times the complexity of a Lipschitz linear function which is computed in
<sup>61</sup> [9], Sec. 26.2. Therefore:

$$R_m(\mathcal{D}_{P,L}) \leq 2K \left\| \begin{bmatrix} c_1/R \\ \boldsymbol{a}_1 \end{bmatrix} \right\|_1$$
$$\times \sup_i \left\| \begin{bmatrix} R \\ \boldsymbol{x}_i \end{bmatrix} \right\|_\infty \sqrt{(2\ln(2d+2))/m}$$
$$\leq 2K \times 2L \times R \times \sqrt{(2\ln(2d+2))/m}.$$

<sup>62</sup>
<sup>63</sup> **A4 PAC bounds for piecewise Bregman divergence metric learning**

<sup>64</sup> In this section we use the Rademacher complexity bounds derived in section A3 along with ap-
<sup>65</sup> proximation guarantees of section A2 to provide standard generalization bounds for empirical risk
<sup>66</sup> minimization under our divergence learning framework.

<sup>67</sup> **Proof of Theorem 2**

<sup>68</sup> The proof is very similar to that of Radamacher complexity bounds for soft-SVM given in [9]. First
<sup>69</sup> from Theorem 26.12 in [9] for a $\rho$-Lipschitz loss function $L(f, z) \leq M$ with probability of at least
<sup>70</sup> $1 - \delta$ we have for all $f \in \mathcal{F}$:

$$L_\mu(f) \leq L_{S_m}(f) + 2\rho R_m(\mathcal{F}) + M\sqrt{(2\ln(2/\delta))/m}.$$

Now note that the hinge loss is 1-Lipschitz, bounded by 1. By substituting $\mathcal{F} = \mathcal{D}_{P,L}$, $f = h_m$ and $L = L^{hinge}$ we get:

$$L_\mu^{hinge}(D_{h_m}) \leq L_{S_m}^{hinge}(D_{h_m}) + 4R_m(\mathcal{D}_{P,L})$$
$$+ \sqrt{(2\ln{(2/\delta)})/m} \quad w.p. \geq 1 - \delta. \tag{4}$$

Since we are also learning the Lipschitz constant $L$, for having a generalization bound we should express a uniform result for all $L$. We use the trick used in [9] for providing the union bound. To proceed for any integer $i$ take $L_i = 2^i$ and take $\delta_i = \delta/(2i^2)$. Using (4) we have for any $L \leq L_i$,

$$L_\mu^{hinge}(D_{h_m}) \leq L_{S_m}^{hinge}(D_{h_m}) + 4R_m(\mathcal{D}_{P,L})$$
$$+ \sqrt{(2\ln{(2/\delta_i)})/m} \quad w.p. \geq 1 - \delta_i.$$

Applying the union bound and noting $\sum_{i=1}^\infty \delta_i \leq \delta$ this holds for all $i$ with probability at least $1 - \delta$. Now take $i = \lceil \log_2 L \rceil \leq \log_2 L + 1$ then $\frac{2}{\delta_i} = \frac{(2i)^2}{\delta} \leq \frac{4\log_2 L}{\delta}$. Therefore:

$$L_\mu^{hinge}(D_{h_m}) \leq L_{S_m}^{hinge}(D_{h_m}) + 4R_m(\mathcal{D}_{P,L})$$
$$+ (\sqrt{4\ln(4\log_2 L) + \ln{(1/\delta)}})/\sqrt{m},$$

with probability at least $1 - \delta$.

## A5 Regression Setting

Next we consider the regression scenario, and discuss generalization bounds. Here we are interested in the expected squared loss between the Bregman divergence obtained from the minimizer of the regression loss (5) and the true divergence value, on unseen (test) data.

Suppose the function $\ell_t$ consists of a pair of points from $X$, say $\boldsymbol{x}_{i_t}$ and $\boldsymbol{x}_{j_t}$, and the $y_t$ value is a noisy version of the the target (ground truth) Bregman divergence between $\boldsymbol{x}_{i_t}$ and $\boldsymbol{x}_{j_t}$. A standard least squares loss function (with no regularization) would seek to solve

$$\min_{\phi \in \mathcal{F}} \sum_{t=1}^m (D_\phi(\boldsymbol{x}_{i_t}, \boldsymbol{x}_{j_t}) - y_t)^2.$$

Suppose we observe the data $S_m = \{(\boldsymbol{x}_{i_t}, \boldsymbol{x}_{j_t}, y_t) | t \in [m]\}$, where $\boldsymbol{x} \in \mathbb{R}^d$ and $y \in \mathbb{R}$. We will model the response random variable $y$ as a Bregman divergence $D_h(\boldsymbol{x}_i, \boldsymbol{x}_j)$ with $h \in \mathcal{F}_{P,L}$. Let $h_m : \mathbb{R}^d \to \mathbb{R}$ be the empirical risk minimizer of

$$\min_{h \in \mathcal{F}_{P,L}} \frac{1}{m} \sum_{t=1}^m (D_h(\boldsymbol{x}_{i_t}, \boldsymbol{x}_{j_t}) - y_t)^2. \tag{5}$$

We know from (4) in the main paper that $D_h(\boldsymbol{x}_i, \boldsymbol{x}_j) = b_i - b_j - \boldsymbol{a}_j^T(\boldsymbol{x}_i - \boldsymbol{x}_j)$, subject to the constraints given in Lemma 1 of the main paper. Therefore (5) can be solved as a quadratic program.

For the following generalization error bounds, we require that the training data be drawn iid. Note that while there are known methods to relax these assumptions, as shown for Mahalanobis metric learning in Bellet and Habrard [2], we assume here for simplicity that data is drawn iid[1] from $\mathcal{X} \times \mathcal{X} \times \mathcal{Y}$ (and analogously for the relative distance case) with distribution $\mu$. Each instance, $t \in [m]$, is a triple, $(\boldsymbol{x}_{i_t}, \boldsymbol{x}_{j_t}, y_t)$ drawn iid from $\mu$.

We have the following result:

**Theorem 1.** *Consider* $S_m = \{(\boldsymbol{x}_{i_t}, \boldsymbol{x}_{j_t}, y_t), t \in [m]\} \sim \mu^m$. *Let* $\| \cdot \|_\mu^2 = \mathbb{E}[| \cdot |^2]$ *and assume,*

$\mathbf{A_1}$: $\|\boldsymbol{x}\|_\infty \leq R$ *and* $\sup |y_t - \mathbb{E}[y_t | \boldsymbol{x}_{i_t}, \boldsymbol{x}_{j_t}]| \leq \sigma$, *i.e. both the input and noise are bounded.*

100     $\mathbf{A_2}$: $E[y_i|\boldsymbol{x}_{i_t}, \boldsymbol{x}_{j_t}] = D_\phi(\boldsymbol{x}_{i_t}, \boldsymbol{x}_{j_t})$, *for a L-Lipschitz $\beta$-smooth function $\phi$.*

101     *The generalization error of the empirical risk minimizer $D_{h_m}$ of the regression loss on $S_m$,*

$$
\begin{aligned}
\|D_{h_m} - y_t\|_\mu^2 \quad \leq \quad & \|D_{h_m} - y_t\|_{S_m}^2 \\
+ \quad & 16MKLR\sqrt{2\ln(2d+2)/m} \\
+ \quad & M^2\sqrt{\ln(1/\delta)/(2m)},
\end{aligned}
$$

102     *with probability at least $1-\delta$. Furthermore, $D_{h_m}$ converges to the ground truth Bregman divergence*
103     *$D_\phi$ and the approximation error is bounded by*

$$
\begin{aligned}
\|D_{h_m} - D_\phi\|_\mu^2 \quad \leq \quad & 36^2\beta^2 R^4 K^{\frac{-2}{d}} \\
+ \quad & 16MKLR\sqrt{2\ln(2d+2)/m} \\
+ \quad & M^2\sqrt{2\ln(2/\delta)/m},
\end{aligned}
$$

104     *where $M = 4LR + \sigma$. By choosing $K = \lceil m^{\frac{d}{4+2d}}\rceil$ we get: $\|D_{h_m} - D_\phi\|_\mu^2 = \mathcal{O}_p(m^{-\frac{1}{d+2}})$.*

105     Consider $S_m \sim \mu^m$ be a set of $m$ i.i.d data points. If $|f(\boldsymbol{x}) - y| \leq M$ for all $f \in \mathcal{F}, \boldsymbol{x}$ and $y$, by a
106     standard Rademacher generalization result:

$$
\|f(\boldsymbol{x}) - y\|_\mu^2 \leq \|f(\boldsymbol{x}) - y\|_{S_m}^2 + 2MR_m(\mathcal{F}) + M^2\sqrt{\frac{\ln 1/\delta}{2m}},
$$

107     with probability greater than $1 - \delta$. By substituting $f = D_{h_m}$, and $\mathcal{F} = \mathcal{D}_{P,L}$ in the above we
108     immediately get the first line of the proposition.

109     Further for the empirical risk minimizer $f_m$ we have that for all $\hat{f} \in \mathcal{F}$ that doesn't depend on the
110     training data $S_m$:

$$
\begin{aligned}
\|f_m(\boldsymbol{x}) - f_*(\boldsymbol{x})\|_\mu^2 \leq \|\hat{f}(\boldsymbol{x}) - f_*(\boldsymbol{x})\|_\mu^2 + 2MR_m(\mathcal{F}) \\
+ 2M^2\sqrt{(\ln 2/\delta)/(2m)},
\end{aligned} \tag{6}
$$

111     where $f_*$ is $\mathbb{E}[y|\boldsymbol{x}]$. This comes from the fact that during training $f_m$ was chosen and not $\hat{f}$. By
112     substituting $f_m = D_{h_m}, f_* = D_\phi, \hat{f} = D_h = \arg\inf_{h \in \mathcal{F}_{P,L}} \|D_\phi - D_h\|_\infty$ and $\mathcal{F} = \mathcal{D}_{P,L}$ in (6)
113     we have:

$$
\begin{aligned}
\|D_{h_m} - D_\phi\|_\mu^2 \leq & \|D_h - D_\phi\|_\mu^2 + 2MR_m(\mathcal{D}_{P,L}) \\
& + 2M^2\sqrt{(\ln 2/\delta)/(2m)} \\
\leq & \|D_h - D_\phi\|_\infty^2 + 2MR_m(\mathcal{D}_{P,L}) \\
& + 2M^2\sqrt{(\ln 2/\delta)/(2m)} \\
\overset{Thm1}{\leq} & (36R^2\beta K^{\frac{-1}{d}})^2 + 2MR_m(\mathcal{D}_{P,L}) \\
& + 2M^2\sqrt{(\ln 2/\delta)/(2m)}.
\end{aligned}
$$

114     Now by substituting $M = 4LR + \sigma$ and $R_m(\mathcal{D}_{P,L})$ from the value given by Lemma 2 we get the
115     proposition. The only thing left to prove is to show $\forall h \in \mathcal{F}_{P,L}$ and $\forall(\boldsymbol{x}, \boldsymbol{x}', y)$; the error is bounded,
116     i.e. $|y - D_h(\boldsymbol{x}, \boldsymbol{x}')| \leq M = 4LR + \sigma$:

$$
\begin{aligned}
|y - D_h| &\leq |D_h - \mathbb{E}[y|\boldsymbol{x}, \boldsymbol{x}']| + |y - \mathbb{E}[y|\boldsymbol{x}, \boldsymbol{x}']| \\
&\leq |D_h - D_\phi| + \sigma \\
&\leq \max\{|D_h|, |D_\phi|\} + \sigma \\
&= |\phi(\boldsymbol{x}) - \phi(\boldsymbol{x}') - \nabla\phi(\boldsymbol{x}')^T(\boldsymbol{x} - \boldsymbol{x}')| + \sigma \\
&\leq 2\|\nabla\phi(\boldsymbol{x}')\|_1\|\boldsymbol{x} - \boldsymbol{x}'\|_\infty + \sigma = 4LR + \sigma.
\end{aligned}
$$

117     **A6 Farthest-point clustering and $K < n$**

118     The algorithm given in the paper assumes that the number of hyperplanes $K$ is equal to $n$; this is
119     mainly for simplicity of presentation. In practice we often want to have $K < n$. Here we discuss
120     details of this approach, which we utilize in our experiments.

We apply a farthest-point clustering to the data first into $K$ clusters, and then fix the assignments of points to hyperplanes using this clustering. With this assignment in place, we can then apply a minor modification to the PBDL algorithm to approximate the Bregman divergence. Farthest-point clustering is a simple greedy algorithm for a K-center problem, where the objective is to divide the space into $K$ partitions such that the farthest distance between a data point and its closest partition center $\mu_i$ is minimized. This problem can be formulated as: given a set of $n$ points $x_1, \ldots, x_n$ a distance metric $\|\cdot\|$ and a predefined partition size $K$, find a partition of data $C_1, \ldots, C_k$ and partition centers $\mu_1, \ldots, \mu_K$ to minimize the maximum radius of the clusters:

$$\max_i \max_{x \in C_i} \|x - \mu_i\|.$$

The farthest point clustering introduced in [5] initially picks a random point $x_{0_0}$ as the center of the first cluster and adds it to the center set $C$. Then for iterations $t = 2$ to $k$ does the following: at iteration $t$, computes the distance of all points from the center set $d(x, C) = \min_{\mu \in C} \|x - \mu\|$. Add the point that has the largest distance from the center set (say $x_{t_0}$) to the center set. Report $x_{0_0}, \ldots, x_{K_0}$ as the partition centers and assign each data point to its closest center.

Authors of [5] proved that farthest-point clustering is a 2-approximation algorithm (i.e. , it computes a partition with maximum radius at most twice the optimum) for any metric. Therefore there is a relation between the partition found by farthest-point clustering and covering set. Assume a set $\{x_1, \ldots, x_n\} \subset \Omega$ has a $\epsilon$-cover of size $K$ over a metric $\|\cdot\|$. The partition found by farthest point clustering of size $K$ is a $2\epsilon$-cover for $\{x_1, \ldots, x_n\}$.

## A7 Parameterizing Bregman divergences by piecewise linear functions

We parameterize the Bregman divergence using max-affine functions $h(x) = \max_{k=1,\ldots,K} a_k^T x + b_k$. Using Lemma 1 from our paper with a predefined partition of the training data points $x_1, \ldots, x_n$ to $\mathcal{C} = \{C_1, \ldots, C_K\}$ and defining the mapping $p_i \doteq k$ given $x_i \in C_k$, we can write any pairwise divergence on training set as

$$\begin{aligned}
D_h(x_i, x_j) &= h(x_i) - h(x_j) - \nabla h(x_j)^T (x_i - x_j) \\
&= (a_{p_i}^T x_i + b_{p_i}) - (a_{p_j}^T x_j + b_{p_j}) - a_{p_j}^T (x_i - x_j) \\
&= b_{p_i} - b_{p_j} + (a_{p_i} - a_{p_j})^T x_i,
\end{aligned}$$

which is linear in terms of the parameters $a_k, b_k, k = 1, \ldots, K$. Therefore if the loss function

$$\mathcal{L}(\phi) = \sum_{i=1}^m c_i(D_\phi, X, y) + \lambda r(\phi),$$

is a convex function of pairwise divergences, it will be a convex loss in terms of parameters. Furthermore one needs to satisfy the constraints given by Lemma 1 in our paper to make sure $h(x)$ remains convex, i.e:

$$b_{p_j} + a_{p_j}^T x_j \geq b_k + a_k^T x_j, \quad j = 1, \ldots, n, \quad k = 1 \ldots, K,$$

which are linear inequality constraints. Therefore one can minimize the loss $\mathcal{L}(\phi)$ as a convex optimization problem.

## A8 Additional Experimental Results

**Bregman divergence regression on synthetic data**

In this section, we experiment with regression tasks on synthetic data. In particular, we show that if data arises from a particular Bregman divergence, our method can discover the underlying divergence whereas Mahalanobis metric learning methods cannot.

**Data**: We generate 100 synthetic data points in three ways: i) discrete probability distributions $\{(p_1, p_2)\}|p_1 + p_2 = 1, p_1, p_2 \geq 0\}$ sampled from a Dirichlet probability distribution $Dir([1]_{1 \times 2})$, with a target value $y$ computed as the *KL* divergence between pairs of distributions; ii) symmetric 2-2 matrices sampled from a Wishart distribution $W_2([1]_{1 \times 2}, 10)$ with target value $y$ computed as the *LogDet* divergence between pairs; iii) data points are sampled uniformly from a unit-ball $\mathcal{B}([0.6]_{1 \times 2}, 1))$ with target value $y$ computed as the *Itakura-Saito* distance between pairs; iv) data points are sampled uniformly from a unit-ball $\mathcal{B}([0.6]_{1 \times 2}, 1))$ with target value $y$ computed as the

Figure 3: Regression with data from various Bregman divergences using PBDL and linear metric learning.

*Mahalanobis* distance between pairs. In each case we add Gaussian noise with stdev 0.05 to the ground truth divergences. For training, we provide all pairs of an increasing set of points ($\{(\boldsymbol{x}_i, \boldsymbol{x}_j), y_{i,j}\}$ for $(i, j)$ in the power set of $\{\boldsymbol{x}_1, \ldots, \boldsymbol{x}_m\}$) and the target values $y_i$ as noisy Bregman divergence of those pairs. For testing, we generate 1000 data points from the same distribution and use noiseless Bregman divergences as targets. Results are averaged over 50 runs.

**Details and observations**: For Bregman regression, we choose the Lipschitz constraint of PBDL for regression to be $\infty$ since the result was not sensitive to the choice of $L$. For *Mahalanbis regression* we do gradient descent for optimizing the least-square fit of a general Mahalanobis metric with the observed data which is done until convergence (as the problem is convex). We see from Figure 2 that Mahalanobis metric learning is not flexible enough to model the data coming from the first three divergences, whereas the proposed divergence learning framework *PBDL* is shown to drastically improve the fit and seems to be a consistent estimator as motivated earlier in Theorem 1.

**Nearest neighbor classification and additional data sets**

We also present results on nearest neighbor classification and more data sets. Table 2 gives some additional performance numbers; in particular, we have added two new data sets and shown results of k-nearest neighbor classification.

Table 1: Learning Bregman divergences (PDBL) compared to existing linear and non-linear metric learning approaches on standard UCI benchmarks. PDBL performs first or second among these benchmarks in 22 of 30 comparisons, outperforming all of the other methods. Note that the top two results for each setting are indicated in bold.

| Data-set | Algorithm | Clustering | | Ranking | | KNN ACC |
| | | Rand-Ind % | Purity % | AUC % | Ave-P % | |
|---|---|---|---|---|---|---|
| Iris | PBDL | **94.5 ± 0.8** | **95.6 ± 0.7** | **96.5 ± 0.4** | **93.5 ± 0.7** | 95.3 ± 0.7 |
| | ITML [3] | **96.4 ± 0.8** | **97.0 ± 0.7** | **97.5 ± 0.3** | **95.3 ± 0.5** | **97.4 ± 0.6** |
| | LMNN [10] | 90.0 ± 1.3 | 91.0 ± 1.3 | 94.3 ± 0.6 | 89.9 ± 0.8 | 96.1 ± 0.6 |
| | GB-LMNN [7] | 88.7 ± 1.5 | 89.9 ± 1.5 | 94.0 ± 0.6 | 89.7 ± 0.8 | 95.6 ± 0.6 |
| | GMML [11] | 93.8 ± 0.9 | 94.5 ± 0.9 | 95.7 ± 0.4 | 92.0 ± 0.6 | **96.6 ± 0.5** |
| | Kernel NCA [4] | 89.9 ± 1.3 | 90.3 ± 1.1 | 93.4 ± 0.6 | 88.3 ± 0.9 | 91.8 ± 1.4 |
| Ionosphere | PBDL | 65.2 ± 1.9 | 77.2 ± 1.9 | 65.8 ± 0.8 | 71.1 ± 0.8 | 81.4 ± 1.0 |
| | ITML | **72.2 ± 1.5** | **83.3 ± 1.2** | **71.5 ± 0.7** | **74.6 ± 0.6** | 85.0 ± 1.0 |
| | LMNN | 58.3 ± 1.2 | 70.8 ± 1.2 | 62.2 ± 1.3 | 69.8 ± 0.9 | **87.1 ± 0.8** |
| | GB-LMNN | 58.5 ± 0.9 | 70.9 ± 1.0 | 64.4 ± 1.3 | 71.2 ± 1.0 | **88.3 ± 0.9** |
| | GMML | 61.7 ± 1.8 | 73.9 ± 1.7 | 66.3 ± 0.8 | 71.3 ± 0.6 | 82.3 ± 1.0 |
| | Kernel NCA | **65.4 ± 1.7** | **77.7 ± 1.5** | **68.8 ± 1.1** | **72.0 ± 0.9** | 84.3 ± 1.0 |
| Balance Scale | PBDL | **84.4 ± 0.7** | **87.8 ± 0.5** | **86.0 ± 0.4** | **82.9 ± 0.5** | **91.4 ± 0.4** |
| | ITML | 68.9 ± 0.9 | 77.5 ± 0.7 | **80.1 ± 0.7** | **74.3 ± 0.8** | **90.0 ± 0.6** |
| | LMNN | 69.5 ± 1.8 | 77.0 ± 1.7 | 75.9 ± 1.3 | 70.0 ± 1.2 | 87.4 ± 0.5 |
| | GB-LMNN | 71.4 ± 1.5 | 79.7 ± 1.4 | 78.1 ± 1.1 | 72.2 ± 1.0 | 87.8 ± 0.6 |
| | GMML | **72.9 ± 0.8** | **80.2 ± 0.8** | 79.0 ± 0.4 | 72.8 ± 0.5 | 87.2 ± 0.6 |
| | Kernel NCA | 65.3 ± 1.5 | 73.0 ± 1.6 | 68.7 ± 1.8 | 63.7 ± 1.9 | 79.9 ± 1.7 |
| Wine | PBDL | **83.7 ± 2.9** | **85.0 ± 3.2** | **91.0 ± 0.9** | **86.7 ± 1.2** | **94.3 ± 0.9** |
| | ITML | 82.8 ± 2.6 | 82.5 ± 3.1 | 89.1 ± 1.1 | 84.6 ± 1.4 | 93.8 ± 1.0 |
| | LMNN | 70.0 ± 0.8 | 68.8 ± 1.2 | 82.4 ± 0.8 | 76.2 ± 1.1 | 91.7 ± 0.8 |
| | GB-LMNN | 70.6 ± 0.9 | 69.3 ± 1.4 | 83.7 ± 0.1 | 78.5 ± 1.3 | 93.8 ± 0.8 |
| | GMML | **83.2 ± 2.9** | **81.0 ± 3.2** | **91.0 ± 0.7** | **88.5 ± 0.7** | **96.5 ± 0.8** |
| | Kernel NCA | 70.4 ± 1.3 | 71.3 ± 1.4 | 75.1 ± 0.9 | 67.7 ± 1.1 | 67.5 ± 1.2 |
| Transfusion | PBDL | 57.9 ± 1.2 | 75.9 ± 0.7 | **54.9 ± 0.4** | **68.2 ± 0.6** | **75.7 ± 0.7**s |
| | ITML | **60.2 ± 1.0** | 75.8 ± 0.7 | 54.2 ± 0.4 | 66.4 ± 0.7 | 74.5 ± 0.6 |
| | LMNN | 59.4 ± 1.3 | **76.3 ± 0.6** | 54.0 ± 0.5 | 67.1 ± 0.7 | 75.0 ± 0.7 |
| | GB-LMNN | 58.9 ± 1.2 | **76.3 ± 0.6** | **54.8 ± 0.6** | 67.2 ± 0.7 | 74.1 ± 0.7 |
| | GMML | 59.3 ± 1.3 | **76.6 ± 0.7** | 54.0 ± 0.5 | **67.5 ± 0.7** | **76.1 ± 0.6** |
| | Kernel NCA | **63.7 ± 0.7** | 76.2 ± 0.7 | 52.2 ± 0.8 | 65.7 ± 0.8 | 74.7 ± 0.8 |
| Figure 1 data | PBDL | **98.2 ± 0.3** | **98.6 ± 0.2** | **97.3 ± 0.3** | **95.6 ± 0.5** | **99.1 ± 0.2** |
| | ITML | **76.2 ± 1.6** | **74.9 ± 2.5** | 90.5 ± 0.5 | 83.7 ± 0.5 | 99.0 ± 0.2 |
| | LMNN | 73.4 ± 1.7 | 69.3 ± 2.6 | 90.4 ± 0.7 | 83.2 ± 0.7 | 98.8 ± 0.2 |
| | GB-LMNN | 73.3 ± .5 | 71.3 ± 2.6 | 90.5 ± 0.8 | 83.4 ± 1.0 | **99.2 ± 0.2** |
| | GMML | 73.9 ± 1.8 | 70.8 ± 2.8 | **91.4 ± 0.2** | **84.2 ± 0.3** | 98.9 ± 0.2 |
| | Kernel NCA | 76.5 ± 2.4 | 73.9 ± 3.7 | 90.4 ± 0.6 | 83.9 ± 0.6 | 98.0 ± 0.5 |

## Footnotes

[1]In many cases this is justified. For instance, in estimating quality scores for items, one often has data corresponding to item-item comparisons [8]; for each item, the learner also observes contextual information. The feedback, $y_t$ depends only on the pair $(\boldsymbol{x}_{i_t}, \boldsymbol{x}_{j_t})$, and as such is independent of other comparisons.