[Reviews · NeurIPS 2020]

Review 1

Summary and Contributions: edit: I read the rebuttal and other reviews, my score has not changed. The paper proposes learn an approximation of a Bregman divergence. Every Bregman divergence can be written as a function of a strictly convex differentiable function. The paper proposes to approximate the strictly convex differentiable function by a non-differentiable convex function written as a maximum of different linear (hence convex) functions. The proposed framework is general and can be applied to different types of losses (e.g. quadruplet-wise constraints in the main paper).

Strengths: Although the approach is simple, the theory of the paper is solid. Rademacher complexity/generalization error bounds are studied. Bregman divergence are a general framework that generalize multiple divergences used in the machine learning literature (e.g. squared Euclidean distance, KL divergence etc...). The experimental evaluation is also solid although most baselines were published about a decade ago (or more). The theoretical study is significant and might help future work working on this kind of generalized divergence. It is also relevant to the NeurIPS community.

Weaknesses: The main limitation/weakness of the paper is that it does not consider more recent baselines and/or deep models. It seems that there is an ICML 2020 paper about deep extensions of this (line 265).

Correctness: The empirical methodology seems correct.

Clarity: The paper is well written and easy to understand although I have not checked the supplementary material.

Relation to Prior Work: Yes. The main novelty compared to previous work are the theoretical generalization error bounds.

Reproducibility: Yes

Additional Feedback:


Review 2

Summary and Contributions: The paper studies the problem of approximating arbitrary Bregman divergences. The underlying idea is to approximate the underlying convex function of the divergence by a piecewise linear function (constructed by selecting, for a given example x, the linear function with maximal value in the set of linear functions used for the approximation). The Rademacher complexity of the class of approximated Bregman divergences using piecewise linear functions is derived. Finally, an algorithm to learn an approximated Brgeman divergence in a Metric Learning setting is proposed. This algorithm comes with generalization guarantees and is shown to be competitive on several tasks.

Strengths: - The idea of using piecewise linear functions to approximate Bregman divergences seems original and is of interest to the Metric Learning community but also to a wider audience given the popularity of Bregman divergences. - The proposed algorithm is sound and backed by theoretical and empirical evidence of its interest. - Theoretically, under reasonable assumptions on the convex function under consideration, it is shown that there exists a piecewise linear function that is a tight approximation provided that the number of linear function considered is large enough. - The Radamacher complexity of Bregman divergences parametrized by piecewise linear functions is investigated.

Weaknesses: - There is a gap between the setting studied in the theoretical analysis and the the one considered in the experiments. Indeed, in the theoretical analysis, pairs or quadruplets of points are assumed to be drawn i.i.d. from a joint distribution while, in the experiments, the points themselves are assumed to be drawn i.i.d. (and, thus, the pairs and quadruplets are not i.i.d. anymore). - The generalization guarantees presented in this paper are not directly comparable to standard metric learning results due to the aforementioned i.i.d. issue.

Correctness: The claims, the method, and the empirical methodology appear to be correct.

Clarity: The paper is mostly clear. Nevertheless, some part could be improved: - Line 27: a_k and b_k are never introduced and it is not clear how they can be obtained at this stage. - Line 143: \phi(xi) should be equal to z_i using (3) rather than (2). - The first constraint in Optimization Problem (7) is hard to parse as it is on two lines. - Line 241: Table 4 is in fact Table 1.

Relation to Prior Work: The relation to prior work appears to be clearly discussed.

Reproducibility: Yes

Additional Feedback: 1) The gap between theory and practice is a bit disappoint. The footnotes at the bottom of page 6 gives a good example of a practical setting aligned with the theory and it would have been interesting to design some experiment in this direction. 2) Line 260-264 of the main paper, preliminary experiments for image retrieval are mentioned using deep features as input. The results are deferred to the supplementary. However, I could not find them there. 3) The regression example in the supplementary shows that the proposed approach is indeed more flexible than Mahalanobis metric learning approaches. This is an interesting experiment that should be mentioned in the main paper. 4) The proposed approach performs quite well on the clustering and ranking experiments. However, its results are slightly lower in terms of KNN accuracy. Similarly the results obtained on ionosphere in the supplementary are far less convincing. Is there any explanation to these differences? On the one hand, the paper presents an interesting and sound approach to approximate Bregman divergences using piecewise linear functions. On the other hand, there is a small gap between the settings considered in theory and the one considered in practice. Overall, I think that this paper could be accepted. --- The rebuttal did not convince me that there is no gap between theory and practice and, thus, my comment 1 still holds. My comments 2 and 4 were not addressed.


Review 3

Summary and Contributions: A method for metric learning based on the Bregman divergence is proposed. The metric, i.e., Bregman divergence is adaptively optimized with a given dataset, using a piece-wise linear function. Validity of the approximation is theoretically investigated and its performance is numerically verified.

Strengths: Using the piece-wise linear function, the method can adaptively approximate the generating function of the Bregman divergence, and its performance is comparable or superior with other metric learning methods.

Weaknesses: The approximation ability by the piece-wise linear function basically depends on the number K of linear functions. In theorem 2, the bound of generalization error is increasing with K, which seems to be strange. Is this correct? In experiments, how to determine K is ambiguous.

Correctness: Yes

Clarity: Yes

Relation to Prior Work: Yes

Reproducibility: Yes

Additional Feedback:


Review 4

Summary and Contributions: This paper proposes to fit Bregman divergence for a given dataset where the convex function for generating the divergence is restricted to be a max-affine function. The learning algorithm is formulated as linear programming. The approximation error by the max-affine function and generalization bound for the metric learning problem with the learned Bregman divergence is derived. Experimental results support the effectiveness of the proposed metric learning method compared to classical ones.

Strengths: Approximating a convex function by max-affine functions is reasonable. Approximation and generalization error bounds are derived, which seem sound. Experimental results show relative superiority to conventional methods.

Weaknesses: The problem statement is not satisfactory hence I'm not confident whether I understand the problem and the authors' claim properly. For example, the assumption K=n (even for the sake of simplicity) seems to make the derived bound useless.

Correctness: I followed proofs in supplementary material, and they seem to be correct.

Clarity: There are many unclear points on the formulation and presentation. The optimization problem (7) is ambiguous. What is the input data and what is the pre-determined parameter? I believe a_i are optimized in the problem (7), but it should be clearly stated. If a_i, i=1,..., K is a set of vectors to be optimized, the number of parameters to be optimized grows with the increase of the number of training samples. The relationship between "n" and "m" is not explained. If "m" is the number of constraints and proportional to "n" and K=n, the generalization bound is meaningless. I guess, partly for that reason, it is important to consider K < n case. K=n is an oversimplification. It is stated that in reality, the authors use K smaller than n, but I cannot find an explanation on how to determine an appropriate number K. Symbols are not used in a consistent manner. lines 54 - 56, "Discuss a generalization error bound for metric learning in the Bregman setting of Op(n−1/2), where n is the number of training points; this matches the bound known for the strictly less general Mahalanobis setting". "n" is said to be the number of training points, but it does not appear in the derived generalization bound.

Relation to Prior Work: I'm not very familiar with recent work on metric learning. Classical and representative methods are properly cited and compared with the proposed method. However, I feel cited papers on metric learning methods are relatively old. I understand recent developments on metric learning are concentrated on deep-learning based methods, and nearly linear method like this submission is not actively studies. So, relatively old references do not affect my rating on this paper so much.

Reproducibility: Yes

Additional Feedback: Please explicitly write the optimization objectives in the mathematical programming. 1st term in r.h.s. of the bound in Theorem 2 is strange. "n", "i", and "t" seem to be confused. Are \deltas in Eq.(11) and Theorem 2 the same thing? I guess not. I understand \delta is a standard notation for PAC bounds, but the same symbol shouldn't be used in the manuscript. ------------ My main concerns are the usefulness of the bound (particularly the relationship between n and K) and presentation issues. Confuse caused by misuse of symbols is addressed by the authors' rebuttal. I understand the rationale behind the seemingly useless generalization bound. I like the main idea of leaning Bregman divergence with piece-wise linear approximation, and raised my score.

[Author Response · NeurIPS 2020]

We thank the reviewers for their comments, and address the main concerns below.

<span style="color:blue">Rev 4: Please explicitly write the optimisation objectives in the mathematical programming.</span>

Suppose we observe $S_m = \{(\boldsymbol{x}_{i_t}, \boldsymbol{x}_{j_t}, \boldsymbol{x}_{k_t}, \boldsymbol{x}_{l_t}) \mid t \in [m]\}$, where $D(\boldsymbol{x}_{i_t}, \boldsymbol{x}_{j_t}) \leq D(\boldsymbol{x}_{k_t}, \boldsymbol{x}_{l_t})$ for some unknown
similarity function and $(i_t, j_t, k_t, l_t)$ are indices of objects in a countable set $\mathcal{U}$ (e.g. set of people in a social network).
To model the underlying similarity function $D$, we propose a Bregman divergence of the form:

$$\hat{D}(\boldsymbol{x}_i, \boldsymbol{x}_j) \triangleq \hat{\phi}(\boldsymbol{x}_i) - \hat{\phi}(\boldsymbol{x}_j) - \nabla_* \hat{\phi}(\boldsymbol{x}_j), \tag{1}$$

where $\hat{\phi}(\boldsymbol{x}) \triangleq \max_{i \in \mathcal{U}_m} \boldsymbol{a}_i^T(\boldsymbol{x} - \boldsymbol{x}_i) + z_i$ , $\nabla_*$ is the biggest sub-gradient, $\mathcal{U}_m$ is the set of all observed objects indices
$\mathcal{U}_m \triangleq \cup_{t=1}^m \{i_t, j_t, k_t, l_t\}$ and $\boldsymbol{a}_i$'s and $z_i$'s are the solution to the following linear program:

$$\min_{z_i, \boldsymbol{a}_i, L} \sum_{t=1}^m \max(\zeta_t, 0) + \lambda L$$

$$\text{s.t.} \begin{cases} z_{i_t} - z_{j_t} - \boldsymbol{a}_{j_t}^T(\boldsymbol{x}_{i_t} - \boldsymbol{x}_{j_t}) + z_{l_t} - z_{k_t} + \boldsymbol{a}_{l_t}^T(\boldsymbol{x}_{k_t} - \boldsymbol{x}_{l_t}) \leq \zeta_t - 1 & t \in [m] \\ z_i - z_j \geq \boldsymbol{a}_j^T(\boldsymbol{x}_i - \boldsymbol{x}_j) & i, j \in \mathcal{U}_m \\ \|\boldsymbol{a}_i\|_1 \leq L & i \in \mathcal{U}_m \end{cases}$$

We will be sure to be explicit about the optimization objective in the final version of the paper.

<span style="color:blue">Rev 4 and Rev 2: Notations, Bounds are meaningless for K=n!, Experiment setup is different from the theory</span>

There are some notational typos which will be corrected. First, the $\delta$ in Theorem 1 is just some dummy variable
denoting something small which we'll replace by another symbol. Second, $n$ stands for number of unique people in all
comparisons. $m$ stands for number of comparisons, i.e: $n = \#\mathcal{U}_m$, so $n$ will increase with $m$. Resolving these typos
lets look at Theorem 2.

**Theorem 2.** *Consider* $S_m = \{(\boldsymbol{x}_{i_t}, \boldsymbol{x}_{j_t}, \boldsymbol{x}_{k_t}, \boldsymbol{x}_{l_t}, y_t), t \in [m]\} \sim \mu^m$, *where* $D(\boldsymbol{x}_{i_t}, \boldsymbol{x}_{j_t}) \leq D(\boldsymbol{x}_{k_t}, \boldsymbol{x}_{l_t})$. *Set*
$R = \max_i \|\boldsymbol{x}_i\|_\infty$. *The generalization error of the learned divergence in (1) when using $K$ hyper-planes satisfies*

$$\mathbb{E}\big[1[\hat{D}(\boldsymbol{x}_{i_t}, \boldsymbol{x}_{j_t}) \geq \hat{D}(\boldsymbol{x}_{k_t}, \boldsymbol{x}_{l_t})]\big] \quad \leq \quad \frac{1}{m} \sum_{t=1}^m \max\big(0, 1 + \hat{D}(\boldsymbol{x}_{i_t}, \boldsymbol{x}_{j_t}) - \hat{D}(\boldsymbol{x}_{k_t}, \boldsymbol{x}_{l_t})\big)$$

$$+ \quad 32KLR\sqrt{2\ln(2d+2)}/\sqrt{m} + \sqrt{4\ln(4\log_2 L) + \ln(1/\delta)}/\sqrt{m},$$

*with probability at least $1 - \delta$ for receiving the data $S_m$.*

**Case 1: (K<n)** We discuss details of the algorithm about the case where $K < n$ in appendix A6; this is another
approach we have used in our experiments which yielded similar results. Using standard cross-validation to select K is
the simplest and most effective way to select a value of K, and also ensures that the theoretical bounds are applicable.
Also its almost obvious that doing further cross validation to choose $K$ would result in improvement over the choice of
$K = n$. However we liked to use $K = n$ in the reported experiments as it results in a faster algorithm and reduces the
time needed for cross validation.

**Case 2: (K=n)** For the theoretical bound to hold we need $n << m$. This could be true in the example of the social
network if we extract some kind of similarity information between people. Regardless of this we found acceptable
results in our experiments with this setting.

**i.i.d setup:** The i.i.d setup is correct in how we trained the Bergman divergence in our experiments. We randomly
choose the similarity comparisons from the fixed classification data-set $\{\boldsymbol{x}_i, y_i\}_{i=1}^n$. This makes sense in a practical
sense too as having or computing relative comparisons between all triplets would make $m = O(n^3)$ which is impractical
when $n$ is large. However we test the divergence in different tasks (i.e. ranking and clustering).

<span style="color:blue">Rev 3: In theorem 2, the bound of generalization error is increasing with K, which seems to be strange. Is this correct?</span>

Yes, this is due a to a bias variance trade-off which is more visible in the regression setting. Increasing $K$ will increase
the variance and worsens the generalisation gap, however it could lower the empirical risk. In experiments, we have
both used cross-validation to choose $K$ as well as using the simpler $K = n$.

[Meta-Review · NeurIPS 2020]

After substantial discussions and an increase of the overall score, the reviewers converged to an overall positive agreement on the paper. On the positive side, the reviewers recognise that the paper is of interest to the community (R2), the algorithm is backed up by experiments and theory (R2), and the rebuttal made a reasonable job of explaining notations ambiguity in generalization bounds (R4). On the negative side however, the reviewers still point a gap between theory and experiments (R2) and maybe a lack of common ground with respect to classical metric learning papers (R2), non-intuitive dependences on some parameters (R3). In the discussions, R1 rallied to R2 in the fact that there is somewhat a gap between theory and experiments. The authors have done a substantial job in explaining notations in their rebuttal and I personally thank them for that; it is strongly recommended they finely polish the camera ready version of the paper, not just with respect to those notations, but also to make clear the gap that needs to be narrowed, maybe in a future work section.